# OpenReview forum: "Behavioral Mode Discovery for Fine-tuning Multimodal Generative Policies"
_ICML.cc/2026/Conference — ICML 2026 regular_

### Official Review · Reviewer_qqPd · 2026-03-12

**Soundness:** 3
**Presentation:** 4
**Significance:** 3
**Originality:** 3
**Overall Recommendation:** 4
**Confidence:** 3

**Summary:**

The submission proposes BMD (Behavioral Mode Discovery), a framework to prevent mode collapse during RL fine-tuning (RLFT) of generative policies. By maximizing the mutual information between latent noise and induced trajectories, BMD uncovers and regularizes latent behavioral modes inherited from demonstrations. This allows the policy to improve task performance while maintaining the multimodal diversity acquired during supervised pre-training.

**Compliance With Llm Reviewing Policy:**

Affirmed.

**Final Justification:**

The rebuttal clarifies the scope of the method, but the core limitation remains unchanged. I maintain my original score.

**Key Questions For Authors:**

1.	Beyond-Demonstration Exploration: Does the BMD regularization prevent the agent from discovering novel, high-performing modes that were not present in the pre-training dataset? Could the authors provide an experiment where the optimal solution requires a behavior qualitatively different from any of the discovered latent modes?

2.	Sensitivity to Latent Prior: How does the choice of the discrete prior $p(z)$ affect the agent's ability to adapt to downstream tasks? If $Z$ is set to a small value, does it force functionally distinct expert behaviors to merge into a single mode?

**Limitations:**

yes

**Strengths And Weaknesses:**

Strengths

1.	The authors strive to discuss the problem of mode collapse in RLFT by providing a solid theoretical link between latent noise and trajectory-level multimodality.

2.	Overall, this study's notable concept concerns the effective use of a steering-policy reparameterization to maintain diverse strategies in complex tasks like Franka Kitchen and ANYmal.

Weakness

mode creation limitation: Overall, an important concept outlined by the article is the preservation of modes already present in the pre-trained generative policy. However, the reviewer thinks the proposed method is structurally limited to mode preservation rather than mode discovery. Since the BMD objective strongly regularizes the policy to stay within the latent modes discovered from the demonstration distribution, it may inherently restrict the RL agent’s ability to explore and acquire genuinely new strategies not present in the initial data (“top grasp" when only "left/right" were demonstrated). In this sense, the framework treats RL more as a tool for fine-tuning existing behaviors rather than a mechanism for expanding the policy's behavioral repertoire beyond the demonstration support.

---

> ### Author Rebuttal · Authors · 2026-03-30
>
> We thank the reviewer for their time and for identifying the solid theoretical grounding linking latent noise to trajectory-level multimodality, the effectiveness of the steering-policy reparameterization across complex benchmarks, and the excellent quality of the presentation.
>
> We also acknowledge the reviewer’s observation regarding mode creation: BMD is intentionally designed for mode preservation during RL fine-tuning, rather than unconstrained skill discovery from scratch. Here, “discovery” refers to uncovering and indexing latent behavioral modes already encoded in a pretrained policy but not explicitly annotated, so that RL fine-tuning can preserve them instead of collapsing to a single behavior. This focus is deliberate: in RLFT, the primary failure mode is not lack of behavioral repertoire, but loss of multimodality under reward optimization.
>
> ### **Q1: Does BMD regularization prevent the discovery of novel, out-of-distribution modes?**
>
>
> BMD does not structurally forbid behaviors outside the demonstration support. The environment reward remains active throughout fine-tuning, and the steering policy can, in principle, move the policy toward behaviors not explicitly represented in the demonstrations when the downstream reward provides a signal for them. Our 2D Gaussian experiment is a simple example of this regime (Figure 2): the downstream reward favors targets at rotated locations relative to the demonstration distribution, and fine-tuning with BMD successfully adapts to these shifted solutions while retaining multimodality.
>
> The harder setting raised by the reviewer is when the downstream optimum requires a qualitatively new behavior that is not represented in the pretrained policy and is also difficult to reach through reward-driven adaptation alone. In that case, BMD is limited by design because its regularizer is meant to preserve discovered pretrained modes rather than introduce an explicit novelty-seeking signal. We view this as an important limitation rather than a contradiction of the method’s goal. Combining BMD with exploration-driven objectives to expand the behavioral modes beyond the pretrained support is an exciting future direction to explore.
>
> ### **Q2: Sensitivity to the latent prior**
>
> The choice of the dimensionality $K$ of the latent space affects adaptation in two distinct ways. First, if $K$ is too small, the model lacks sufficient capacity to represent all recoverable behavioral modes, so some of them must necessarily be merged or omitted, as the reviewer suggested. Second, even when $K$ is large enough in principle, optimization under the variational MI objective may still allocate multiple latent codes to the same high-reward behavior, leaving other valid modes underrepresented during fine-tuning. This effect is illustrated in the ANYmal locomotion visualization (Fig. 13c, Appendix J.2), where one of the codes collapses onto the same locomotion direction while other behaviorally distinct directions are not retained.
>
> Appendix H.3 suggests that mild overparameterization is an effective practical remedy: choosing $K$ somewhat larger than the expected number of dominant modes provides sufficient representational flexibility, and in practice redundant codes tend to collapse benignly to existing behaviors rather than destabilizing training.

---

> > ### Author Rebuttal · Reviewer_qqPd · 2026-04-04
> >
> > We thank the authors for the clear and thoughtful responses. The rebuttal effectively clarifies the intended scope of the method and addresses my concern regarding mode discovery by positioning BMD as a mode-preserving approach rather than a discovery mechanism. This improves the overall clarity and understanding of the contribution.
> >
> > However, the fundamental limitation remains unchanged, as the method is inherently designed for preserving existing modes rather than expanding the behavioral repertoire beyond the demonstration support. As a result, the overall scope and impact of the work are similar to my original assessment.
> >
> > Accordingly, I will maintain my original score.

---

### Official Review · Reviewer_xX9u · 2026-03-12

**Soundness:** 3
**Presentation:** 3
**Significance:** 3
**Originality:** 4
**Overall Recommendation:** 4
**Confidence:** 4

**Summary:**

Authors prospose a way to obtain multi-modal policy via steering diffusion policy. PPO (or any other RL algorithm) is attached with an new intrinsic reward that includes a term from variational approximation of mutual information between latent code z and state s. Latent codes will index modes.

**Compliance With Llm Reviewing Policy:**

Affirmed.

**Final Justification:**

I thank authors for their diligent rebuttal. I feel that paper could be further strengthened by explicit MI consistency checks and theoretical analysis of Q3. I will keep my original score of 4.

**Key Questions For Authors:**

- In line 232 next state is sampled from the forward model. Above it says that "rollout the policy", how to reconcile this?
- Isn't Eq. (1) (Barber-Agakov, 2003) MI estimator? When taking supremum over q() one will obtain MI exactly.
- Is there interference between lines 240 and 242 (policy training and q())? Rollouts are taken from policy and policy depends on q() via intrinsic reward. Can this result into an oscillation behavior?

**Limitations:**

-

**Strengths And Weaknesses:**

Soundness:
Strengths: System as proposed is solid. All elements individually do make sense.
Weaknesses: How good is the variational MI estimator? Obviously, accurate MI estimation is not necessary, but self-consistency checks (as proposed by Song and Ermon, 2020) should be applied.

Presentation: Paper as it is written is clear. But some looseness or inaccuracies exists in mathematical presentation. Such as conditional MI in line 163 is incorrect, whereas in Appendix D it is correct. Goal in line 121 appears to be combination of two losses, which raises the question of interaction of these gradients. But ultimately, authors use intrinsic reward formulation.

Significance: Results are modarately significant.

Originality: Ideas, however combination of existing ideas are a result of novel combination.

---

> ### Author Rebuttal · Authors · 2026-03-30
>
> We thank the reviewer for the careful reading, for recognizing the technical soundness of the overall system, and for highlighting the originality of the paper as a principled combination of ideas.
>
> We also appreciate the reviewer’s point regarding variational MI estimation and self-consistency checks. This is a valid concern in general: variational MI estimators are known to exhibit bias/variance issues and can fail basic self-consistency properties when interpreted as accurate estimators of the true MI. In our case, however, Eq. (1) is used primarily as a training proxy for mode distinguishability rather than as an absolute estimate of mutual information. Consistent with this role, we empirically validate that the resulting quantity is monotonic with the number of modes in a controlled setting (Table 1), and we evaluate it through its downstream effect on multimodal retention. We agree that adding self-consistency checks could further strengthen the estimator-focused discussion, and we will clarify in the revision that our goal is not exact MI estimation but a practical and stable intrinsic signal for preserving pretrained modes.
>
> We also thank the reviewer for reporting the typo in line 163, we will fix this in the revision.
>
>
> ### **Q1: Line 232 — next state sampled from forward model vs. "rollout the policy"**
>
> Thank you for pointing this out. By “rollout the policy” we mean executing the current policy in the environment: at each step, the policy samples $a_t \sim \pi(\cdot \mid s_t, z)$, and the environment then produces $s_{t+1} \sim p(\cdot \mid s_t, a_t)$. Thus, the rollout is generated jointly by the policy and the environment dynamics. Our use of “forward model” was only meant to denote the simulator transition function, not a learned dynamics model; we will revise this wording for clarity.
>
> ### **Q2: Is Eq. (1) the Barber-Agakov (2003) MI estimator?**
>
> Yes. Equation 1 is the Barber-Agakov variational lower bound, which becomes tight when $q(z \mid s)$ equals the true posterior $p(z \mid s)$. We cite Eysenbach et al. (2019) [1] for the derivation in the context of skill discovery, as it is not our main contribution, but the original bound is due to Barber-Agakov.
>
> ### **Q3: Oscillation between policy updates and $q_\phi$ updates?**
>
> Thank you for raising this point. Yes, this type of interference is well known in the skill-discovery literature: because rollouts are generated by the current policy, while the policy is itself updated using an intrinsic reward induced by $q_\phi$, the joint optimization can, in principle, introduce non-stationarity and oscillatory behavior. In our setting, however, this issue is mitigated for three reasons: (1) we start from a pretrained policy that already exhibits multimodal structure, rather than discovering skills from scratch; (2) we pretrain the inference model and steering policy on the intrinsic objective before introducing the environment reward; and (3) we use alternating updates, so $q_\phi$ is first updated on the current rollout batch and the policy is then updated using the resulting intrinsic reward. Empirically, we observe stable training across tasks and seeds, and Appendix J.1 shows a smooth trade-off between task reward and multimodality rather than oscillatory dynamics.
>
> [1] Eysenbach, B., Gupta, A., Ibarz, J. and Levine, S., 2018. Diversity is all you need: Learning skills without a reward function. ICLR 2019.

---

> > ### Author Rebuttal · Reviewer_xX9u · 2026-04-03
> >
> > I thank authors for their diligent rebuttal. I feel that paper could be further strengthened by explicit MI consistency checks and theoretical analysis of Q3.

---

### Official Review · Reviewer_Hj3y · 2026-03-13

**Soundness:** 3
**Presentation:** 3
**Significance:** 3
**Originality:** 3
**Overall Recommendation:** 4
**Confidence:** 3

**Summary:**

The paper investigates the reinforcement learning (RL) fine-tuning of pre-trained generative policies, focusing on maximizing task performance while maintaining multimodal behavior. To prevent the behavioral collapse typical of standard RL fine-tuning, the authors propose behavioral mode discovery (BMD), an unsupervised mode-discovery framework. This method utilizes a mutual-information proxy for multimodality and identifies latent behavioral modes through a combination of a steering policy and an inference model. The resulting intrinsic reward is used to regularize the fine-tuning process. The framework is evaluated across several domains, including a 2D Gaussian-mixture environment, simulated robotic manipulation in ManiSkill, ANYmal locomotion, and the Franka Kitchen benchmark. The results indicate that BMD frequently matches or exceeds the task success of standard fine-tuning baselines while demonstrating superior preservation of behavioral diversity.

**Compliance With Llm Reviewing Policy:**

Affirmed.

**Key Questions For Authors:**

- Since the evaluation of multimodality relies on ground-truth modes in simulation and heuristic assignments in ManiSkill, could the authors clarify the sensitivity of their conclusions to these specific mode definitions?

- Table 5 illustrates significant improvements with BMD on Reach and Lift, yet only partial multimodality recovery on Avoid. Could the authors comment on which task properties determine whether BMD will perform optimally or less effectively?

- Given that the method is built upon a mutual-information proxy and a discrete latent space, could the authors provide insights into how sensitive practical performance is to these specific modeling choices?

**Limitations:**

yes

**Strengths And Weaknesses:**

Strengths

- Significance of the Problem: The paper addresses a critical challenge in generative policy deployment: the tendency of standard RL fine-tuning to optimize for rewards at the expense of the diverse behaviors acquired during supervised pre-training. This motivation is clearly and effectively established in the introduction.

- Methodological Structure: The technical approach is well-organized. Sections 4.1–4.3 provide formal definitions for the mutual-information proxy, the latent mode-discovery mechanism, and the regularized objective. Additionally, Figure 1 provides a helpful visual reference for the overall pipeline.

- Breadth of Empirical Evaluation: The study covers a diverse range of environments, from controlled Gaussian-mixture settings to complex manipulation and locomotion tasks (ManiSkill, ANYmal, Franka Kitchen). The provided tables generally support the claim that BMD effectively preserves multimodality compared to standard baselines.

- Reasonably Broad Baseline Comparison: The evaluation is not restricted to a single family of methods; it includes direct fine-tuning, residual fine-tuning, and steering-policy baselines. The inclusion of BMD-augmented variants further strengthens the informativeness of the empirical results.

Weaknesses

- Dependency on Heuristic Mode Assignments: A portion of the empirical evaluation relies on task-specific or heuristic definitions of "modes." Section 5 notes that simulated evaluations assume access to ground-truth modes, while ManiSkill tasks utilize heuristics to categorize trajectories. This dependency slightly obscures the purity of the diversity measurements.

- Non-Uniform Performance Gains: While the results are generally strong, BMD does not consistently dominate across every method-task combination. Specifically, certain BMD variants exhibit partial behavioral collapse or only marginal improvements in more challenging environments—a limitation acknowledged within the paper itself.

- Sensitivity to Modeling Choices: The primary multimodality metric relies on a mutual-information proxy and a discrete latent space. The paper identifies this as a limitation, yet it remains unclear how sensitive the framework's overall efficacy is to these specific architectural and modeling decisions.

---

> ### Author Rebuttal · Authors · 2026-03-30
>
> We thank the reviewer for their time and for highlighting the significance of the problem, the methodological clarity of the formalization, the breadth of empirical evaluation across diverse domains, and the informative multi-family baseline comparison as key strengths of the work.
>
> ### **Q1: Sensitivity of conclusions to mode definitions**
>
> In ManiSkill, modes are assigned using spatial heuristics (e.g., grasp orientation or spatial location) that reflect clear geometric differences between successful strategies and correctly separate the demonstrations used for pretraining. This is also consistent with prior multimodal behavior benchmarks such as D3IL [1], which explicitly specify distinct valid behaviors and introduce practical metrics to quantify diversity.
>
> Importantly, our claim is primarily relative rather than absolute: all methods are evaluated under the same mode assignment rule, and the observed gap between standard fine-tuning baselines and BMD is large. Therefore, while the exact values of $\mathrm{mc}@0.80$ or $\mathcal{H}$ may vary somewhat with the mode definition, the conclusion that BMD preserves substantially more behavioral diversity is less sensitive to it.
>
> [1] Jia, X., Blessing, D., Jiang, X., Reuss, M., Donat, A., Lioutikov, R. and Neumann, G., 2024. Towards diverse behaviors: A benchmark for imitation learning with human demonstrations. ICLR 2024.
>
>
> ### **Q2: Why does BMD only partially recover modes on Avoid? When does BMD is more or less effective?**
>
> BMD is most effective when successful behaviors are comparably rewarded and can be captured by a relatively simple latent mode structure. It becomes less effective when the task exhibits: (i) a strongly unbalanced reward landscape, which biases RL fine-tuning toward a subset of high-return modes; (ii) mode differences that emerge only later in the trajectory, making them harder to represent with a single latent variable (we discuss in the conclusion how hierarchical mode representations could mitigate this issue); and (iii) a sharper diversity–success trade-off, since preserving lower-reward modes requires a stronger regularization signal.
>
> The Avoid environment is a representative example of this regime: it has 24 modes, strong trajectory-length asymmetry (shorter paths to avoidance goals dominate the reward signal, biasing fine-tuning toward a subset of modes regardless of regularization strength), and behavioral differences that become salient later in execution. As a result, BMD substantially mitigates collapse, but does not fully recover all modes.  The $\lambda$-ablation in Appendix J.1 suggests that increasing $\lambda$ might help retain more modes in the Avoid environment at a modest cost of a lower success rate.
>
>
>
> ### **Q3: Sensitivity to MI proxy and discrete latent space**
>
> Table 1 validates that M from Eq. (1) is monotonically consistent with the true number of
> modes (0.00 → 0.58 → 1.06 for 1, 2, 4 modes), confirming it as a reliable proxy when
> multimodality exists. We note, however, that MI-based objectives are known to reward distinguishability rather than metric spread, and therefore may not always capture the broadest notion of behavioral diversity. For instance, two trajectories that are behaviorally very similar but differ in a small spatial aspect may still be assigned to different modes if that difference is sufficient for discrimination (see Figure 13.c for an example of such a case).
>
> For the discrete latent space, Appendix H.3 shows that performance is reasonably robust to the choice of latent dimensionality $K$ of the latent space. In particular, mild overparameterization is not harmful: when $K$ exceeds the true number of modes, redundant latent values tend to map to the same behavior rather than destabilizing training. Empirically, the key requirement is that
> $K$ should be large enough to represent the dominant modes present in the pretrained policy. Otherwise, the policy might collapse some latents to the same mode as visualized in Figure 13.c in the Appendix.
>
> We chose the discrete formulation because it provides a simple and stable way to index recoverable behaviors, in the spirit of prior skill-discovery methods such as DIAYN [2]. That said, extending the framework to continuous or more structured latent spaces is an interesting direction for future work, especially in settings where the underlying behavioral variation is not naturally discrete.
>
> [2] Eysenbach, B., Gupta, A., Ibarz, J. and Levine, S., 2018. Diversity is all you need: Learning skills without a reward function. ICLR 2019.

---

> > ### Author Rebuttal · Reviewer_Hj3y · 2026-04-04
> >
> > The rebuttal answers my questions and concerns well. I keep my positive recommendation of this paper.

---

### Official Review · Reviewer_u8N7 · 2026-03-17

**Soundness:** 3
**Presentation:** 3
**Significance:** 3
**Originality:** 2
**Overall Recommendation:** 4
**Confidence:** 4

**Summary:**

The paper addresses the problem of behavioral mode collapse during RL fine tuning of pretrained policies. The key idea is discovering modes in the pretrained policy and using diversity as a DIAYN-like intrinsic reward during downstream fine-tuning along with the environment reward. Experiments in multiple sim environments validate that BMD outperforms standard RL fine tuning baselines in preserving mode coverage.

**Compliance With Llm Reviewing Policy:**

Affirmed.

**Final Justification:**

The paper proposes a clean and effective approach to address mode collapse in RL post-training, and the rebuttal response has clarified my concerns.

**Key Questions For Authors:**

- Does it work on vision-based environments?
- Could the authors compare this with baselines that specifically optimize for coverage (see above), in addition to vanilla BC baselines?

**Limitations:**

Yes.

**Strengths And Weaknesses:**

- Mode collapse in RL fine tuning is an important problem.
- The paper is clearly written and easy to follow.
- The idea of using a DIAYN-like mutual information metric to quantify multimodal coverage as an intrinsic reward is well-motivated.
- Empirical evaluations in sim show that BMD outperforms baselines for mode coverage.
- Ablations validate the design and choice of hyperparameters.
- It's somewhat sensitive to the dimensionality of the discrete latent, which is a limitation that the authors also addressed in H.3, and the regularization weight $\lambda$, which is ablated in J.1.
- State-based experiments only. It would be good to see this scale to vision-based experiments.
- My main concern is that the proposed method is only compared to vanilla baselines which do not optimize for mode coverage. It would be good to see a comparison with well-established exploration objectives such as random network distillation, or simply a state entropy maximization term baseline.

---

> ### Author Rebuttal · Authors · 2026-03-30
>
> We thank the reviewer for the thoughtful assessment and for recognizing the importance of the problem, the clarity of the presentation, the motivation for the MI-based objective, and the value of the ablations.
> ### **Q1: Comparison against coverage-oriented baselines (RND, state entropy maximization)**
>
> We appreciate this suggestion and acknowledge that it strengthens the empirical case. We want to first clarify a key conceptual distinction: RND and state entropy maximization encourage visitation of *novel or diverse states*, but they do not condition on specific behavioral modes and therefore cannot guarantee that distinct pre-existing modes are individually preserved and controllable. In contrast, BMD explicitly discovers and indexes behavioral modes via z, enabling the steering policy to selectively maintain each one.
>
> We agree that comparing against objectives explicitly encouraging coverage strengthens the empirical case. Following this suggestion, we evaluated two additional baselines on ANYmal (4 modes): `DSRL[ENTROPY]` , using an APT-style state-entropy bonus, and `DSRL[RND]` , using a novelty bonus based on random network distillation as suggested by the reviewer.
>
>
> **Table — ANYmal locomotion (4 modes). New rows: `DSRL[ENTROPY]`, `DSRL[RND]`.**
>
> | **Method** | $\\mathrm{SR}$ $(\\uparrow)$ | $\\mathrm{SR}_{\\mathrm{M}}$ $(\\uparrow)$ | $\\mathrm{mc}@0.80$ $(\\uparrow)$ | $\\mathcal{H}$ $(\\uparrow)$ |
> |:---|:---:|:---:|:---:|:---:|
> | `PRE` | $0.41 \\scriptscriptstyle\\pm 0.01$ | $0.39 \\scriptscriptstyle\\pm 0.01$ | $0.00 / 4$ | $0.96 \\scriptscriptstyle\\pm 0.00$ |
> | `DSRL` | $1.00 \\scriptscriptstyle\\pm 0.00$ | $0.25 \\scriptscriptstyle\\pm 0.00$ | $1.00 / 4$ | $0.00 \\scriptscriptstyle\\pm 0.00$ |
> | `DSRL[ENTROPY]` | $1.00 \\scriptscriptstyle\\pm 0.00$ | $0.25\\scriptscriptstyle\\pm 0.00$ | $1.00 / 4$ | $0.00 \\scriptscriptstyle\\pm 0.00$ |
> | `DSRL[RND]` | $1.00 \\scriptscriptstyle\\pm 0.00$ | $0.25 \\scriptscriptstyle\\pm 0.00$ | $1.00 / 4$ | $0.00 \\scriptscriptstyle\\pm 0.00$ |
> | `DSRL[BMD]` | $0.97 \\scriptscriptstyle\\pm 0.01$ | $0.89 \\scriptscriptstyle\\pm 0.12$ | $3.67 / 4$ | $0.91 \\scriptscriptstyle\\pm 0.12$ |
>
> Both `DSRL[ENTROPY]` and `DSRL[RND]` collapse to one mode, with $\mathrm{mc}@0.80 = 1.00/4$ and $\mathcal{H}=0.00$, matching unconditioned `DSRL`, whereas `DSRL[BMD]` recovers $3.67/4$ modes with $\mathcal{H}=0.91$.
>
> This behavior is also consistent with prior work on multimodal RL with diffusion policies. DDiffPG [1] uses RND-driven exploration together with an additional mode-aware training mechanism, namely mode-specific Q-learning over multimodal batches, rather than relying on exploration bonuses alone. Their framing likewise suggests that exploration can help discover diversity, but that preserving multiple behaviors under a greedy RL objective requires an explicit mode-aware learning signal. Our setting differs in two important ways: we study RL fine-tuning of pretrained generative policies rather than learning from scratch, and BMD additionally yields controllable mode retention through the latent variable and steering policy.
>
> [1] Li, Z., Krohn, R., Chen, T., Ajay, A., Agrawal, P. and Chalvatzaki, G., 2024. Learning multimodal behaviors from scratch with diffusion policy gradient. Advances in Neural Information Processing Systems, 37, pp.38456-38479.
>
>
> ### **Q2: Vision-based experiments**
>
> We agree that vision-based evaluation is important. We note, however, that extending BMD to images is primarily a representation learning problem, not an issue with the objective itself. BMD requires a representation from which modes are identifiable; in state-based settings this is given, while in image-based settings it must be learned. This is consistent with recent unsupervised RL work such as METRA[2], which argues that scaling diversity-based objectives to pixels requires learning compact latent abstractions with behaviorally meaningful geometry rather than operating directly in raw observation space.
>
> A practical extension of BMD in this direction is to use asymmetric fine-tuning: the policy operates from images, while the inference model estimating the MI regularizer receives privileged state information. This isolates the representation learning issue from the core contribution and provides a straightforward path to vision-based RL fine-tuning. A fully image-based version of BMD would instead require learning a visual representation that makes behavioral modes identifiable from observations alone.
>
> While this work focuses on establishing the connection between unsupervised skill discovery and multimodal fine-tuning of generative policies, and on showing that this connection helps preserve multimodal behaviors, extending the framework to vision-based policies and exploring alternative skill-discovery objectives are important directions for future work.
>
> [2] Park, S., Rybkin, O. and Levine, S., 2023. Metra: Scalable unsupervised rl with metric-aware abstraction. ICLR 2024.

---

> > ### Author Rebuttal · Reviewer_u8N7 · 2026-04-05
> >
> > Thank you for the clarifications and additional experiments, and I stand corrected. Overall the approach makes sense and I think this is a solid paper.

---

### Decision · Program_Chairs · 2026-04-30

**Decision:**

Accept (regular)

**Comment:**

This paper proposes Behavioral Mode Discovery (BMD), a framework for RL fine-tuning of pretrained generative policies that preserves multimodal behavioral diversity by discovering latent modes and using a mutual information-based intrinsic reward as a regularizer. The work addresses the well-recognized problem of mode collapse during RL fine-tuning and establishes a principled connection between unsupervised skill discovery and generative policy fine-tuning.

All four reviewers recommend weak accept. The reviewers recognized the clear motivation and presentation, the solid theoretical grounding linking latent noise to trajectory-level multimodality, and the breadth of empirical evaluation across diverse environments (ANYmal locomotion, ManiSkill, Franka Kitchen). The multi-family baseline comparison and ablations were also appreciated.

The main concerns raised during review were: (1) the absence of comparisons against coverage-oriented baselines such as RND or state entropy maximization; (2) the lack of vision-based experiments; (3) the accuracy and self-consistency of the variational MI estimator; (4) the potential for oscillatory behavior between policy and inference model updates; and (5) the inherent limitation that BMD is designed for mode preservation rather than the discovery of genuinely new out-of-distribution behaviors.

The authors addressed most of these concerns satisfactorily in the rebuttal with several experiment and clarifications Two reviewers marked their concerns as fully resolved; the other two noted that explicit MI self-consistency checks and a more thorough theoretical treatment of training stability would further strengthen the paper, but did not raise their concerns to a blocking level.

I believe this paper makes a solid and timely contribution to the area of generative policy fine-tuning, and I recommend weak accept.